# PyGlove: Symbolic Programming
# for Automated Machine Learning

**Daiyi Peng, Xuanyi Dong, Esteban Real, Mingxing Tan, Yifeng Lu**
**Hanxiao Liu, Gabriel Bender, Adam Kraft, Chen Liang, Quoc V. Le**
Google Research, Brain Team

{daiyip, ereal, tanmingxing, yifenglu,
hanxiaol, gbender, adamkraft, crazydonkey, qvl}@google.com
xuanyi.dxy@gmail.com [*]

## Abstract

Neural networks are sensitive to hyper-parameter and architecture choices. Auto-mated Machine Learning (AutoML) is a promising paradigm for automating these choices. Current ML software libraries, however, are quite limited in handling the dynamic interactions among the components of AutoML. For example, efficient NAS algorithms, such as ENAS [1] and DARTS [2], typically require an implementation coupling between the search space and search algorithm, the two key components in AutoML. Furthermore, implementing a complex search flow, such as searching architectures within a loop of searching hardware configurations, is difficult. To summarize, changing the search space, search algorithm, or search flow in current ML libraries usually requires a significant change in the program logic.

In this paper, we introduce a new way of programming AutoML based on symbolic programming. Under this paradigm, ML programs are mutable, thus can be manipulated easily by another program. As a result, AutoML can be reformulated as an automated process of symbolic manipulation. With this formulation, we decouple the triangle of the search algorithm, the search space and the child program. This decoupling makes it easy to change the search space and search algorithm (without and with weight sharing), as well as to add search capabilities to existing code and implement complex search flows. We then introduce PyGlove, a new Python library that implements this paradigm. Through case studies on ImageNet and NAS-Bench-101, we show that with PyGlove users can easily convert a static program into a search space, quickly iterate on the search spaces and search algorithms, and craft complex search flows to achieve better results.

## 1 Introduction

Neural networks are sensitive to architecture and hyper-parameter choices [3, 4]. For example, on the ImageNet dataset [5], we have observed a large increase in accuracy thanks to changes in architectures, hyper-parameters, and training algorithms, from the seminal work of AlexNet [5] to recent state-of-the-art models such as EfficientNet [6]. However, as neural networks become increasingly complex, the potential number of architecture and hyper-parameter choices becomes numerous. Hand-crafting neural network architectures and selecting the right hyper-parameters is, therefore, increasingly difficult and often take months of experimentation.

Automated Machine Learning (AutoML) is a promising paradigm for tackling this difficulty. In AutoML, selecting architectures and hyper-parameters is formulated as a search problem, where a *search space* is defined to represent all possible choices and a *search algorithm* is used to find the

---

[*]Work done as a research intern at Google.

best choices. For hyper-parameter search, the search space would specify the range of values to try. For architecture search, the search space would specify the architectural configurations to try. The search space plays a critical role in the success of neural architecture search (NAS) [7, 8], and can be significantly different from one application to another [8–11]. In addition, there are also many different search algorithms, such as random search [12], Bayesian optimization [13], RL-based methods [1, 9, 14, 15], evolutionary methods [16], gradient-based methods [2, 10, 17] and neural predictors [18].

This proliferation of search spaces and search algorithms in AutoML makes it difficult to program with existing software libraries. In particular, a common problem of current libraries is that search spaces and search algorithms are tightly coupled, making it hard to modify search space or search algorithm alone. A practical scenario that arises is the need to upgrade a search algorithm while keeping the rest of the infrastructure the same. For example, recent years have seen a transition from AutoML algorithms that train each model from scratch [8, 9] to those that employ weight-sharing to attain massive efficiency gains, such as ENAS and DARTS [1, 2, 14, 15, 19]. Yet, upgrading an existing search space by introducing weight-sharing requires significant changes to both the search algorithm and the model building logic, as we will see in Section 2.2. Such coupling between search spaces and search algorithms, and the resulting inflexibility, impose a heavy burden on AutoML researchers and practitioners.

We believe that the main challenge lies in the programming paradigm mismatch between existing software libraries and AutoML. Most existing libraries are built on the premise of immutable programs, where a fixed program is used to process different data. On the contrary, AutoML requires programs (i.e. model architectures) to be mutable, as they must be dynamically modified by another program (i.e. the search algorithm) whose job is to explore the search space. Due to this mismatch, predefined interfaces for search spaces and search algorithms struggle to accommodate unanticipated interactions, making it difficult to try new AutoML approaches. *Symbolic programming*, which originated from LISP [20], provides a potential solution to this problem, by allowing a program to manipulate its own components as if they were plain data [21]. However, despite its long history, symbolic programming has not yet been widely explored in the ML community.

In this paper, we reformulate AutoML as an automated process of manipulating ML programs symbolically. Under this formulation, programs are mutable objects which can be cloned and modified after their creation. These mutable objects can express standard machine learning concepts, from a convolutional unit to a complex user-defined training procedure. As a result, all parts of a ML program are mutable. Moreover, through symbolic programming, programs can modify programs. Therefore the interactions between the child program, search space, and search algorithm are no longer static. We can mediate them or change them via meta-programs. For example, we can map the search space into an abstract view which is understood by the search algorithm, translating an architectural search space into a super-network that can be optimized by efficient NAS algorithms.

Further, we propose *PyGlove*, a library that enables general symbolic programming in Python, as an implementation of our method tested on real-world AutoML scenarios. With PyGlove, Python classes and functions can be made mutable through brief Python annotations, which makes it much easier to write AutoML programs. PyGlove allows AutoML techniques to be easily dropped into preexisting ML pipelines, while also benefiting open-ended research which requires extreme flexibility.

To summarize, our contributions are the following:

- We reformulate AutoML under the symbolic programming paradigm, greatly simplifying the programming interface for AutoML by accommodating unanticipated interactions among the child programs, search spaces and search algorithms via a mutable object model.
- We introduce *PyGlove*, a general symbolic programming library for Python which implements our symbolic formulation of AutoML. With PyGlove, AutoML can be easily dropped into preexisting ML programs, with all program parts searchable, permitting rapid exploration on different dimensions of AutoML.
- Through case studies, we demonstrate the expressiveness of PyGlove in real-world search spaces. We demonstrate how PyGlove allows AutoML researchers and practitioners to change search spaces, search algorithms and search flows with only a few lines of code.

## 2   Symbolic Programming for AutoML

Many AutoML approaches (e.g., [2, 9, 22]) can be formulated as three interacting components: the *child program*, the *search space*, and the *search algorithm*. AutoML's goal is to discover a performant child program (e.g., a neural network architecture or a data augmentation policy) out of a large set

of possibilities defined by the search space. The search algorithm accomplishes the said goal by iteratively sampling child programs from the search space. Each sampled child program is then evaluated, resulting in a numeric measure of its quality. This measure is called the *reward*[2]. The reward is then fed back to the search algorithm to improve future sampling of child programs.

In typical AutoML libraries [23–31], these three components are usually tightly coupled. The coupling between these components means that we cannot change the interactions between them unless non-trivial modifications are made. This limits the flexibility of the libraries. Some successful attempts have been made to break these couplings. For example, Vizier [26] decouples the search space and the search algorithm by using a dictionary as the search space contract between the child program and the search algorithm, resulting in modular black-box search algorithms. Another example is the NNI library [27], which tries to unify search algorithms with and without weight sharing by carefully designed APIs. This paper, however, solves the coupling problem in a different and more general way: with symbolic programming, programs are allowed to be modified by other programs. Therefore, instead of solving fixed couplings, we allow dynamic couplings through a mutable object model. In this section, we will explain our method and show how this makes AutoML programming more flexible.

## 2.1 AutoML as an Automated Symbolic Manipulation Process

AutoML can be interpreted as an automated process of searching for a child program from a search space to maximize a reward. We decompose this process into a sequence of symbolic operations. A (regular) child program (Figure 1-a) is *symbolized* into a symbolic child program (Figure 1-b), which can be then cloned and modified. The symbolic program is further *hyperified* into a search space (Figure 1-c) by replacing some of the fixed parts with to-be-determined specifications. During the search, the search space is *materialized* into different child programs (Figure 1-d) based on search algorithm decisions, or can be rewritten into a super-program (Figure 1-e) to apply complex search algorithms such as efficient NAS.

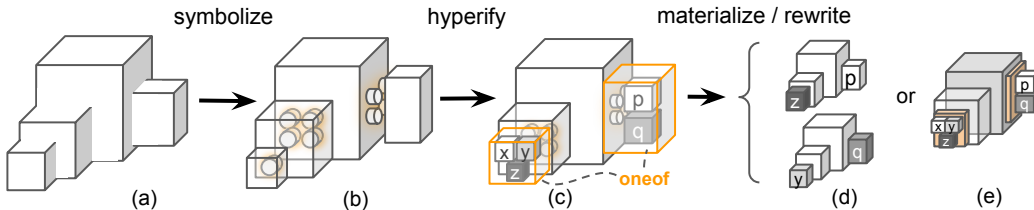

Figure 1: AutoML as an automated symbolic manipulation process.

An analogy to this process is to have a robot build a house with LEGO [32] bricks to meet a human being's taste: symbolizing a regular program is like converting molded plastic parts into LEGO bricks; hyperifying a symbolic program into a search space is like providing a blueprint of the house with variations. With the help of the search algorithm, the search space is materialized into different child programs whose rewards are fed back to the search algorithm to improve future sampling, like a robot trying different ways to build the house and gradually learning what humans prefer.

**Symbolization.** A (regular) child program can be described as a complex object, which is a composition of its sub-objects. A symbolic child program is such a composition whose sub-objects are no longer tied together forever, but are detachable from each other hence can be replaced by other sub-objects. The symbolic object can be hierarchical, forming a *symbolic tree* which can be *manipulated* or *executed*. A symbolic object is manipulated through its hyper-parameters, which are like the studs of a LEGO brick, interfacing connections with other bricks. However, symbolic objects, unlike LEGO bricks, can have internal states which are automatically recomputed upon modifications. For

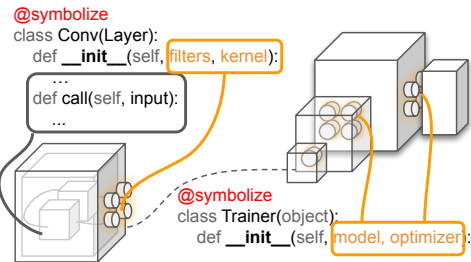

Figure 2: Symbolizing classes into mutable symbolic trees. Their hyper-parameters are like the studs of LEGO bricks, while their implementations are less interesting while we manipulate the trees.

example, when we change the dataset of a trainer, the train steps will be recomputed from the number of examples in the dataset if the training is based on the number of epochs. With such a mutable object model, we no longer need to create objects from scratch repeatedly, or modify the producers up-stream, but can clone existing objects and modify them into new ones. The symbolic tree representation puts an emphasis on manipulating the object definitions, while leaving the implementation details behind. Figure 2 illustrates the symbolization process.

## 2.2 Disentangling AutoML through Symbolic Programming

**Disentangling search spaces from child programs.** The search space can be disentangled from the child program in that 1) the classes and functions of the child program can be implemented without depending on any AutoML library (Appendix B.1.1), which applies to most preexisting ML projects whose programs were started without taking AutoML in mind; 2) a child program can be manipulated into a search space without modifying its implementation. Figure 3 shows that a child program is turned into a search space by replacing a *fixed* Conv with a *choice* of Identity, MaxPool and Conv with searchable filter size. Meanwhile, it swaps a *fixed* Adam optimizer with a *choice* between the Adam and an RMSProp with a searchable learning rate.

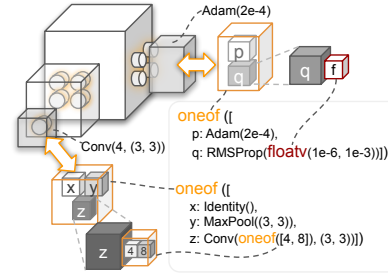

Figure 3: Hyperifying a child program into a search space by replacing fixed parts with to-be-determined specifications.

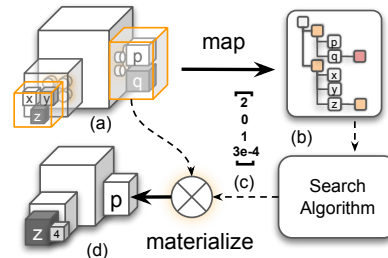

Figure 4: Materializing a (concrete) child program (d) from the search space (a) with an abstract child program (c) proposed from the search algorithm, which holds an abstract search space (b) as the algorithm's view for the (concrete) search space.

**Disentangling search spaces from search algorithms.** Symbolic programming breaks the coupling between the search space and the search algorithm by preventing the algorithm from seeing the full search space specification. Instead, the algorithm only sees what it needs to see for the purposes of searching. We refer to the algorithm's view of the search space as the *abstract search space*. The full specification, in contrast, will be called the *concrete search space* (or just the "search space" outside this section). The distinction between the concrete and abstract search space is illustrated in Figure 4: the concrete search space acts as a boilerplate for producing concrete child programs, which holds all the program details (e.g., the fixed parts). However, the abstract search space only sees the parts that need decisions, along with their numeric ranges. Based on the abstract search space, an *abstract child program* is proposed, which can be static numeric values or variables. The static form is for obtaining a concrete child program, shown in Figure 4, while the variable form is used for making a super-program used in efficient NAS – the variables can be either discrete for RL-based use cases or real-valued vectors for gradient-based methods. Mediated by the abstract search space and the abstract child program, the search algorithm can be thoroughly decoupled from the child program. Figure 5 gives a more detailed illustration of Figure 4.

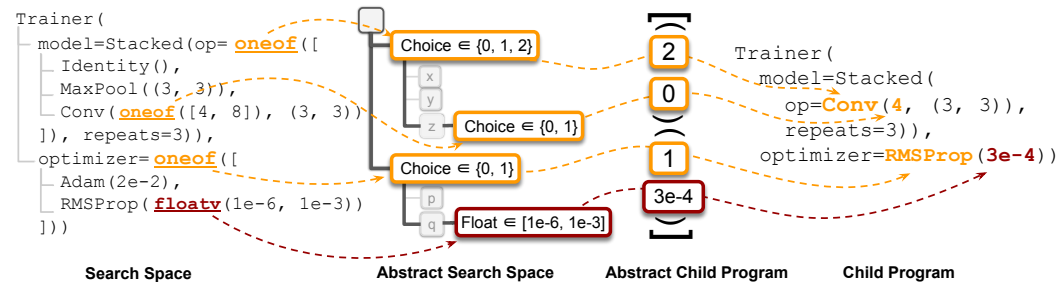

Figure 5: The path from a (concrete) search space to a (concrete) child program. The disentanglement between the search space and the search algorithm is achieved by (1) abstracting the search space, (2) proposing an abstract child program, and (3) materializing the abstract child program into a concrete one.

**Disentangling search algorithms from child programs.**
While many search algorithms can be implemented by
rewriting symbolic objects, complex algorithms such as
ENAS [1], DARTS [2] and TuNAS [15] can be decom-
posed into 1) a child-program-agnostic algorithm, plus 2) a
meta-program (e.g. a Python function) which rewrites the
search space into a representation required by the search al-
gorithm. The meta-program only manipulates the symbols
which are interesting to the search algorithm and ignores
the rest. In this way, we can decouple the search algorithm
from the child program.

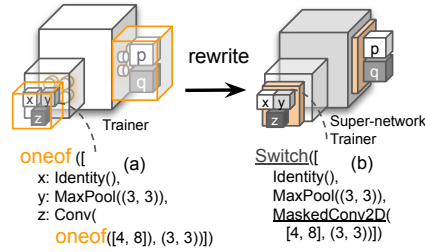

Figure 6: Rewriting a search space (a) into
a super-program (b) required by TuNAS.

For example, the TuNAS [15] algorithm can be decomposed into 1) an implementation of RE-
INFORCE [33] and 2) a rewrite function which transforms the architecture search space into a
super-network, and replaces the regular trainer with a trainer that samples and trains the super-
network, illustrated in Figure 6. If we want to switch the search algorithm to DARTS [2], we use a
different rewrite function that generates a super-network with soft choices, and replace the trainer
with a super-network trainer that updates the choice weights based on the gradients.

## 2.3 Search space partitioning and complex search flows

Early work [19, 34, 35] shows that factorized search can help partition the computation for optimizing
different parts of the program. Yet, complex search flows have been less explored, possibly due
in part to their implementation complexity. The effort involved in partitioning a search space and
coordinating the search algorithms is usually non-trivial. However, the symbolic tree representation
makes search space partitioning a much easier task: with a partition function, we can divide those
to-be-determined parts into different groups and optimize each group separately. As a result, each
optimization process sees only a portion of the search space – a sub-space – and they work together
to optimize the complete search space. Section 3.4 discusses common patterns of such collaboration
and how we express complex search flows.

# 3 AutoML with PyGlove

In this section, we introduce PyGlove, a general symbolic programming library on Python, which
also implements our method for AutoML. With examples, we demonstrate how a regular program
is made symbolically programmable, then turned into search spaces, searched with different search
algorithms and flows in a dozen lines of code.

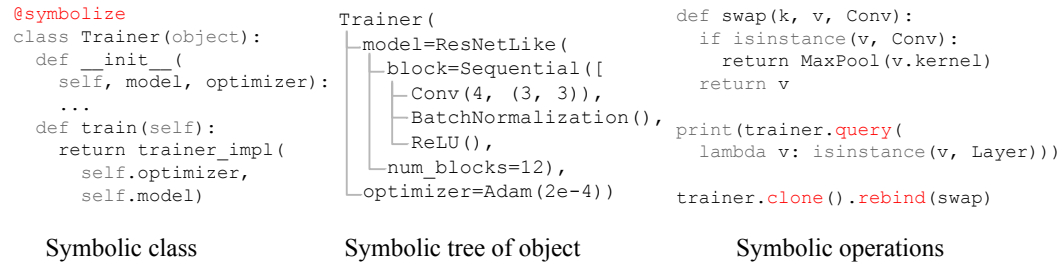

Figure 7: A regular Python class made symbolically programmable via the `symbolize` decorator (left), whose
object is a symbolic tree (middle), in which all nodes can be symbolically operated (right). For example, we
can (i) retrieve all the `Layer` objects in the tree via query, (ii) `clone` the object and (iii) modify the copy by
swapping all `Conv` layers with `MaxPool` layers of the same kernel size using `rebind`.

## 3.1 Symbolize a Python program

In PyGlove, preexisting Python programs can be
made symbolically programmable with a *sym-
bolize* decorator. Besides classes, functions
can be symbolized too, as discussed in Ap-
pendix B.1.2. To facilitate manipulation, Py-
Glove provides a wide range of symbolic opera-
tions. Among them, *query*, *clone* and *rebind* are of special importance as they are foundational to

Table 1: The development cost of dropping PyGlove
into existing projects on different ML frameworks. The
source code of MNIST is included in Appendix B.5.

| Projects | Original lines of code | Modified lines of code |
|---|---|---|
| PyTorch ResNet [36] | 353 | 15 |
| TensorFlow MNIST [37] | 120 | 24 |

other symbolic operations. Examples of these operations can be found in Appendix B.2. Figure 7 shows (1) a symbolic Python class, (2) an instance of the class as a symbolic tree, and (3) key symbolic operations which are applicable to a symbolic object. To convey the amount of work required to drop PyGlove into real-life projects, we show the number of lines of code in making a PyTorch [36] and a TensorFlow [37] projects searchable in Table 1.

## 3.2 From a symbolic program to a search space

With a child program being a symbolic tree, *any node* in the tree can be replaced with a to-be-determined specification, which we call *hyper value* (in correspondence to *hyperify*, a verb introduced in Section 2.1 in making search spaces). A search space is naturally represented as a symbolic tree with hyper values. In PyGlove, there are three classes of hyper values: 1) a continuous value declared by `floatv`; 2) a discrete value declared by `intv`; and 3) a categorical value declared by `oneof`, `manyof` or `permutate`. Table 2 summarizes different hyper value classes with their semantics. Figure 8 shows a search space that jointly optimizes a model and an optimizer. The model space is a number of blocks whose structure is a sequence of permutation from [`Conv`, `BatchNormalization`, `ReLU`] with searchable filter size.

*Dependent hyper-parameters* can be achieved by using higher-order symbolic objects. For example, if we want to search for the filters of a `Conv`, which follows another `Conv` whose filters are twice the input filters, we can create a symbolic `Block` class, which takes only one filter size – the output filters of the first `Conv` – as its hyper-parameters. When it's called, it returns a sequence of 2 `Conv` units based on its filters, as shown in Figure 9. The filters of the block can be a hyper value at construction time, appearing as a node in the symbolic tree, but will be materialized when it's called.

```
Trainer(
├─model=ResNetLike(
│  ├─block=Sequential(
│  │  └─permutate([
│  │     ├─Conv(
│  │     │  ├─filters=oneof([4, 8]),
│  │     │  └─kernel=(3, 3)),
│  │     ├─BatchNormalization(),
│  │     └─ReLU()
│  │     ])),
│  └─num_blocks=intv(6, 12)),
└─optimizer=oneof([
   ├─Adam(2e-4),
   └─RMSProp(floatv(1e-6, 1e-3))
   ]))
```

Figure 8: The child program from Figure 7-2 is turned into a search space.

```
@symbolize
class Block(object):

  def __init__(self, filters):
    self.filters = filters

  def __call__(self):
    return Sequential([
      Conv(self.filters, (3, 3)),
      Conv(self.filters*2, (3, 3))])
```

Figure 9: Expressing dependent hyper-parameters by introducing a higher-order symbolic `Block` class.

## 3.3 Search algorithms

Without interacting with the child program and the search space directly, the search algorithm in PyGlove repeatedly 1) proposes an abstract child program based on the abstract search space and 2) receives measured qualities for the abstract child program to improve future proposals. PyGlove implements many search algorithms, including Random Search, PPO and Regularized Evolution.

Table 2: Hyper value classes and their semantics.

| Strategy | Hyper-parameter annotation | Search space semantics |
|---|---|---|
| Continuous | `floatv`(min, max) | A float value from $\mathbb{R}^{[min, max]}$ |
| Discrete | `intv`(min, max) | An int value from $\mathbb{Z}^{[min, max]}$ |
| Categorical | `oneof`(candidates) | Choose 1 out of N candidates |
| | `manyof`(K, candidates, $\theta$) | Choose K out of N candidates with optional constraints $\theta$ on the uniqueness and order of chosen candidates |
| | `permutate`(candidates) | A special case of `manyof` which searches for a permutation of all candidates |
| Hierarchical | (when a categorical hyper value contains child hyper values) | Conditional search space |

## 3.4 Expressing search flows

With a search space, a search algorithm, and an optional search space partition function, a search flow can be expressed as a for-loop, illustrated in Figure 10-left. Search space partitioning enables various ways in optimizing the divided sub-spaces, resulting in three basic search types: 1) optimize the sub-

```
for trainer, feedback in sample(
    search_space=hyper_trainer,
    algorithm=PPO(),
    partition_fn=None):
  reward = trainer.train()
  feedback(reward)
```

| Search type | for-loop pattern |
|---|---|
| Joint | $\mathtt{for}(x, f_x) : ...$ |
| Separate | $\mathtt{for}(x_1, f_{x1}) : ...$ <br> $\mathtt{for}(x_2, f_{x2}) : ...$ |
| Factorized | $\mathtt{for}(x_1, f_{x1}) :$ <br> $\quad\mathtt{for}(x_2, f_{x2}) : ...$ |

Figure 10: PyGlove expresses search as a for-loop (left). Complex search flows can be expressed as compositions of for-loops (right).

spaces *jointly*; 2) optimize the sub-spaces *separately*; or 3) *factorize* the optimization. Figure 10-right maps the three search types into different compositions of for-loop.

Let's take the search space defined in Figure 8 as an example, which has a hyper-parameter sub-space (the hyper `optimizer`) and an architectural sub-space (the hyper `model`). Towards the two sub-spaces, we can 1) jointly optimize them without specifying a partition function, as is shown in Figure 10-left; 2) separately optimize them, by searching the hyper `optimizer` first with a fixed `model`, then use the best optimizer found to optimize the hyper `model`; or 3) factorize the optimization, by searching the hyper `optimizer` with a partition function in the outer loop. Each example in the loop is a trainer with a fixed `optimizer` and a hyper `model`; the latter will be optimized in the inner loop. The combination of these basic patterns can express very complex search flows, which will be further studied through our NAS-Bench-101 experiments discussed in Section 4.3.

### 3.5 Switching between search spaces

Making changes to the search space is a daily routine for AutoML practitioners, who may move from one search space to another, or to combine orthogonal search spaces into more complex ones. For example, we may start by searching for different operations at each layer, then try the idea of searching for different output filters (Figure 11), and eventually end up with searching for both. We showcase such search space exploration in Section 4.2.

```
def relax_filters(k, v, parent):
  if isinstance(parent, Conv):
    if k == 'filters':
      return oneof([v//2, v, v*2])
  return v

hyper_trainer = trainer.clone()
  .rebind(relax_filters)
```

Figure 11: Manipulating the model in a trainer into a search space by relaxing the fixed filters of the `Conv` as a set of options.

### 3.6 Switching between search algorithms

The search algorithm is another dimension to experiment with. We can easily switch between search algorithms by passing a different algorithm to the `sample` function shown in Figure 10-1. When applying efficient NAS algorithms, the `hyper_trainer` will be rewritten into a trainer that samples and trains the super-network transformed from the architectural search space.

## 4 Case Study

In this section, we demonstrate that with PyGlove how users can define complex search spaces, explore new search spaces, search algorithms, and search flows with simplicity.

### 4.1 Expressing complex search spaces

The composition of hyper values can represent complex search spaces. We have reproduced popular NAS papers, including NAS-Bench-101 [38], MNASNet [8], NAS-FPN [39], ProxylessNAS [14], TuNAS [15], and NATS-Bench [40]. Here we use the search spaces from NAS-Bench-101, NAS-FPN, and TuNAS to demonstrate the expressiveness of PyGlove.

In the NAS-Bench-101 search space (Figure 12-top), there are $N$ different positions in the network and $\binom{N}{2} = \frac{N(N-1)}{2}$ edge positions that can be independently turned on or off. Each node independently selects one of $K$ possible operations.

```
# NAS-Bench-101
ModelSpec(
  nodes=[oneof(range(K))]*N,
  edges=[oneof([0, 1])]*N*(N-1)/2)

# NAS-FPN
FpnNode(
  type=oneof(['sum', 'attention']),
  level=3,
  input_offsets=manyof(
      2, range(NUM_PRE_NODES),
      distinct=True,
      sorted=True))

# TuNAS
Residual(oneof([
  InvertedBottleneck(
    filters=oneof([32, 48, 64]),
    kernel=oneof([3, 5, 7]),
    expansion=oneof([3, 6])),
  Zero()]))
```

Figure 12: Partial search space definition for NAS-Bench-101 (top), NAS-FPN (middle) and TuNAS (bottom).

The NAS-FPN search space is a repeated FPN cell, each of whose nodes (Figure 12-middle) aggregates two outputs of previous nodes. The aggregation is either sum or global attention. We use `manyof` with the constraints *distinct* and *sorted* to select input nodes without duplication.

The TuNAS search space is a stack of blocks, each containing a number of residual layers (Figure 12-bottom) of inverted bottleneck units, whose filter size, kernel size and expansion factor will be tuned. To search the number of layers in a block, we put `Zeros` as a candidate in the `Residual` layer so the residual layer may downgrade into an identity mapping.

## 4.2  Exploring search spaces and search algorithms

We use MobileNetV2 [41] as an example to demonstrate how to explore new search spaces and search algorithms. For a fair comparison, we first retrain the MobileNetV2 model on ImageNet to obtain a baseline. With our training setup, it achieves a validation accuracy of 73.1% (Table 3, row 1) compared with 72.0% in the original MobileNetV2 paper. Details about our experiment setup, search space definitions, and the code for creating search spaces can be found in Appendix C.1.

**Search space exploration:** Similar to previous AutoML works [8, 14], we explore 3 search spaces derived from MobileNetV2 that tune the hyper-parameters of the inverted bottleneck units [41]: (1) Search space $\mathcal{S}_1$ tunes the kernel size and expansion ratio. (2) Search space $\mathcal{S}_2$ tunes the output filters (3) Search space $\mathcal{S}_3$ combines $\mathcal{S}_1$ and $\mathcal{S}_2$ to tune the kernel size, expansion ratio and output filters.

From Table 3, we can see that with PyGlove we were able to convert MobileNetV2 into $\mathcal{S}_1$ with *23 lines* of code (row 2) and $\mathcal{S}_2$ with *10 lines of code* (row 5). From $\mathcal{S}_1$ and $\mathcal{S}_2$, we obtain $\mathcal{S}_3$ in just *a single line of code* (row 6) using `rebind` with chaining the transform functions from $\mathcal{S}_1$ and $\mathcal{S}_2$.

**Search algorithm exploration:** On the search algorithm dimension, we start by exploring different search algorithms on $\mathcal{S}_1$ using black-box search algorithms (Random Search [12], Bayesian [26]) and then efficient NAS (TuNAS [15]). To make model sizes comparable, we constrain the search to 300M multiply-adds[3] using TuNAS's absolute reward function [15]. To switch between these algorithms, we only had to *change 1 line of code*.

Table 3: Programming cost of switching between *three* search spaces and *three* AutoML algorithms based on PyGlove. Lines of code in red is the cost in creating new search spaces, while the lines of code in black is the cost for switching algorithms. The unit cost for search and training is defined as the TPU hours to train a MobileNetV2 model on ImageNet for 360 epochs. The test accuracies and MAdds are based on 3 runs.

| # | Search space | Search algorithm | **Lines of codes** | Search cost | Train cost | Test accuracy | # of MAdds |
|---|---|---|---|---|---|---|---|
| 1 | (*static*) | N/A | N/A | N/A | 1 | $73.1 \pm 0.1$ | 300M |
| 2 | (*static*) $\to \mathcal{S}_1$ | RS | **+23** | 25 | 1 | $73.7 \pm 0.3$ ($\uparrow 0.6$) | $300 \pm 3$ M |
| 3 | $\mathcal{S}_1$ | RS $\to$ Bayesian | **+1** | 25 | 1 | $73.9 \pm 0.3$ ($\uparrow 0.8$) | $301 \pm 5$ M |
| 4 | $\mathcal{S}_1$ | Bayesian $\to$ TuNAS | **+1** | 1 | 1 | $74.2 \pm 0.1$ ($\uparrow 1.1$) | $301 \pm 5$ M |
| 5 | (*static*) $\to \mathcal{S}_2$ | TuNAS | **+10** | 1 | 1 | $73.3 \pm 0.1$ ($\uparrow 0.2$) | $302 \pm 7$M |
| 6 | $\mathcal{S}_1, \mathcal{S}_2 \to \mathcal{S}_3$ | TuNAS | **+1** | 2 | 1 | $73.8 \pm 0.1$ ($\uparrow 0.7$) | $302 \pm 6$M |

## 4.3  Exploring complex search flows on NAS-Bench-101

PyGlove can greatly reduce the engineering cost when exploring complex search flows. In this section, we explore various ways to optimize the NAS-Bench-101 search space. NAS-Bench-101 is a NAS benchmark where the goal is to find high-performing image classifiers in a search space of neural network architectures. This search space requires optimizing both the types of neural network layers used in the model (e.g., 3x3 Conv) and how the layers are connected.

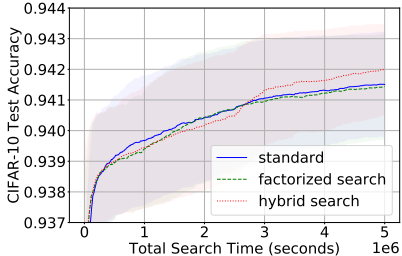

Figure 13: Mean and standard deviation of search performances with different search flows on NAS-Bench-101 (500 runs), using Regularized Evolution [16] .

We experiment with *three* search flows in this exploration: 1) we reproduce the original paper to establish a baseline, which uses the search space defined in Figure 12-top to *jointly* optimize the nodes and edges. 2) we try a *factorized* search, which optimizes the nodes in the outer loop and the edges in the inner loop – the reward for a node setting is computed as the average

of top 5 rewards from the architectures sampled in the inner loop. While its performance is not as good as the baseline under the same search budget, we suspect that under each fixed node setting, the edge space is not explored enough. 3) To alleviate this problem, we come out a *hybrid* solution, which uses the first half of the budget to optimize the nodes as in search flow 2, while using the other half to optimize the edges, based on the best node setting found in the first phase. Interestingly, the search trajectory crosses over the baseline in the second phase, ended with a noticeable margin (Figure 13). We used Regularized Evolution [16] for all these searches, each with 500 runs. It takes only *15 lines of code* to implement the factorized search and *26 lines of code* to implement the hybrid search. Source codes are included in Appendix C.2.

## 5    Related Work

**Software frameworks**  have greatly influenced and fueled the advancement of machine learning. The need for computing gradients has made auto-gradient based frameworks [36, 37, 42–45] flourish. To support modular machine learning programs with the flexibility to modify them, frameworks were introduced with an emphasis on hyper-parameter management [46, 47]. The sensitivity of machine learning to hyper-parameters and model architecture has led to the advent of AutoML libraries [23–31]. Some (e.g., [23–25]) formulate AutoML as a problem of jointly optimizing architectures and hyper-parameters. Others (e.g., [26–28]) focus on providing interfaces for black-box optimization. In particular, Google's Vizier library [26] provides tools for optimizing a user-specified search space using black-box algorithms [12, 48], but makes the end user responsible for translating a point in the search space into a user program. DeepArchitect [29] proposes a language to create a search space as a program that connects user components. Keras-tuner [30] employs a different way to annotate a model into a search space, though this annotation is limited to a list of supported components. Optuna [49] embraces eager evaluation of tunable parameters, making it easy to declare a search space on the go (Appendix B.4). Meanwhile, efficient NAS algorithms [1, 2, 14] brought new challenges to AutoML frameworks, which require coupling between the controller and child program. AutoGluon [28] and NNI [27] partially solve this problem by building predefined modules that work in both general search mode and weight-sharing mode, however, supporting different efficient NAS algorithms are still non-trivial. Among the existing AutoML systems we are aware of, complex search flows are less explored. Compared to existing systems, PyGlove employs a mutable programming model to solve these problems, making AutoML easily accessible to preexisting ML programs. It also accommodates the dynamic interactions among the child programs, search spaces, search algorithms, and search flows to provide the flexibility needed for future AutoML research.

**Symbolic programming** , where a program manipulates symbolic representations, has a long history dating back to LISP [20]. The symbolic representation can be programs as in meta-programming, rules as in logic programming [50] and math expressions as in symbolic computation [51, 52]. In this work, we introduce the symbolic programming paradigm to AutoML by manipulating a symbolic tree-based representation that encodes the key elements of a machine learning program. Such program manipulation is also reminiscent of program synthesis [53–55], which searches for programs to solve different tasks like string and number manipulation [56–59], question answering [60, 61], and learning tasks [62, 63]. Our method also shares similarities with prior works in non-deterministic programming [64–66], which define non-deterministic operators like *choice* in the programming environment that can be connected to optimization algorithms. Last but not least, our work echos the idea of building robust software systems that can cope with unanticipated requirements via advanced symbolic programming [67].

## 6    Conclusion

In this paper, we reformulate AutoML as an automated process of manipulating a ML program through symbolic programming. Under this formulation, the complex interactions between the child program, the search space, and the search algorithm are elegantly disentangled. Complex search flows can be expressed as compositions of for-loops, greatly simplifying the programming interface of AutoML without sacrificing flexibility. This is achieved by resolving the conflict between AutoML's intrinsic requirement in modifying programs and the immutable-program premise of existing software libraries. We then introduce PyGlove, a general-purpose symbolic programming library for Python which implements our method and is tested on real-world AutoML scenarios. With PyGlove, AutoML can be easily dropped into preexisting ML programs, with all program parts searchable, permitting rapid exploration of different dimensions of AutoML.

## Broader Impact

Symbolic programming/PyGlove makes AutoML more accessible to machine learning practitioners, which means manual trial-and-error of many categories can be replaced by machines. This can also greatly increase the productivity of AutoML research, at the cost of increasing demand for computation, and – a result – increasing $CO_2$ emissions.

We see a big potential in symbolic programming/PyGlove in making machine learning researchers more productive. On a new ground of mutable programs, experiments can be reproduced more easily, modified with lower cost, and shared like data. A large variety of experiments can co-exist in a shared code base that makes combining and comparing different techniques more convenient.

Symbolic programming/PyGlove makes it much easier to develop search-based programs which can be used in a broad spectrum of research and product areas. Some potential areas, such as medicine design, have a clear societal benefit, while others potential applications, such as video surveillance, could improve security while raising new privacy concerns.

## Acknowledgments and Disclosure of Funding

We would like to thank Pieter-Jan Kindermans and David Dohan for their help in preparing the case study section of this paper; Jiquan Ngiam, Rishabh Singh for their feedback to the early versions of the paper; Ruoming Pang, Vijay Vasudevan, Da Huang, Ming Cheng, Yanping Huang, Jie Yang, Jinsong Mu for their feedback at early stage of PyGlove; Adams Yu, Daniel Park, Golnaz Ghiasi, Azade Nazi, Thang Luong, Barret Zoph, David So, Daniel De Freitas Adiwardana, Junyang Shen, Lav Rai, Guanhang Wu, Vishy Tirumalashetty, Pengchong Jin, Xianzhi Du, Yeqing Li, Xiaodan Song, Abhanshu Sharma, Cong Li, Mei Chen, Aleksandra Faust, Yingjie Miao, JD Co-Reyes, Kevin Wu, Yanqi Zhang, Berkin Akin, Amir Yazdanbakhsh, Shuyang Cheng, HyoukJoong Lee, Peisheng Li and Barbara Wang for being early adopters of PyGlove and their invaluable feedback.

Funding disclosure: This work was done as a part of the authors' full-time job in Google.

## Footnotes

[2]While we use RL concepts to illustrate the core idea of our method, as will be shown later, the proposed paradigm is applicable to other types of AutoML methods as well.

[3]For RS and Bayesian, we use rejection sampling to ensure sampled architectures have around 300M MAdds.

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
