[Supplementary Material]

# Appendix of PyGlove

**Daiyi Peng, Xuanyi Dong, Esteban Real, Mingxing Tan, Yifeng Lu**
**Hanxiao Liu, Gabriel Bender, Adam Kraft, Chen Liang, Quoc V. Le**
Google Research, Brain Team

{daiyip, ereal, tanmingxing, yifenglu,
hanxiaol, gbender, adamkraft, crazydonkey, qvl}@google.com
xuanyi.dxy@gmail.com [*]

Appendix A provides a formal definition of symbolic programs in our method, including symbolic counterparts of different program constructs, supported operations, and the description of algorithms used in the materialization process. Appendix B gives a more detailed introduction to PyGlove – our implementation of the method – with an example of dropping neural architecture search (NAS) into an existing Tensorflow program (MNIST [1]). Appendix C provides additional information for experiments used in our case studies, including experiment setup, source code for creating search spaces and complex search flows.

## A    More on Symbolic Programming for AutoML

### A.1    Formal definition of a symbolic program

Give a program construct type $t$, let the hyper-parameters (which defines the uniqueness of an instance of $t$) be noted as $P(t) = \langle p_0, ..., p_n \rangle$. The symbolic type of $t$ can then be defined as the output of the symbolization function $S$ applied on $t$, which returns a tuple of $t$'s type information and its hyper-parameter definitions:

$$s = S(t) = \langle t, P(t) \rangle \tag{1}$$

A hyper-parameter $p_i$ of $s$ is either a primitive type or a symbolic type. Therefore an instance $x$ of $s$ – a symbolic object – is a tree node, whose sub-nodes are its hyper-parameters. For convenience, $x$ is called a symbolic $t$, e.g: symbolic `Dataset`, symbolic `Conv`, etc. A symbolic program is a symbolic object that can be executed, for example, a symbolic `Trainer` that trains and evaluates a `ResNet` (as a sub-node) on ImageNet.

Two tree nodes are equal if and only if their type and hyper-parameters are equal. For example, consider a symbolic `Conv` class which takes `filters`, `kernel_size` as its hyper-parameters. Two `Conv` instances are equal if and only if their `filters` and `kernel_size` are equal.

We can clone a tree by copying its type information and hyper-parameters. Similarly, we can replace a hyper-parameter value with a new value, which is the foundation for symbolic manipulation. For example, a symbolic `Conv`'s `kernel_size` can be changed from $(3, 3)$ to $(5, 5)$ by another program.

Symbolic constraints can be specified on the hyper-parameters. These constraints define the hyper-parameters' value types and ranges. When a value is assigned as a hyper-parameter of another symbolic object, it will be validated based on the symbolic constraint on that hyper-parameter. Since the sub-nodes of a symbolic object can be manipulated, the constraints are helpful in catching mistakes during symbolic manipulation.

---

[*]Work done as a research intern at Google.

## A.2 Symbolic types

The basic elements of a computer program are classes and functions, plus a few built-in data structure that works with the classes and functions for composition. To symbolize a computer program, we need to map these basic program constructs to their symbolic counterparts. Based on Equation 1, the symbolic type of $t$ is defined by $t$'s type information and hyper-parameters, illustrated in Table 1

Table 1: Hyper-parameters of basic program constructs

| Program construct type | Hyper-parameters |
|:---:|:---:|
| class | Constructor arguments. |
| function | Function arguments. |
| list | Indices in the list. |
| dict | Keys in the dict. |

Though a regular function takes arguments, the function itself doesn't hold its hyper-parameter values. Therefore, in order to manipulate the hyper-parameters of a function, a symbolic function – functor – behaves like an object: an function with bound arguments. As a result, a functor is no different from a class object with a call method, whose arguments could be bound either at construction time or call time. Therefore, a functor can be a node in the symbolic tree.

## A.3 Operations on symbolic types

Symbolic objects can be manipulated via a set of operations. Table 2 lists the basic operations applicable to all symbolic types. Particularly, `rebind` in the modification category is of special importance, as it's the foundation for implementing complex program transforms.

Table 2: Basic operations applicable to symbolic types.

| Category | Operation | Description |
|:---:|:---:|:---:|
| Modification | $\text{rebind}(x, \text{dict})$ | Replace each node in $x$ whose path is a key in `dict` |
| | $\text{rebind}(x, \lambda)$ | Recursively apply the function $\lambda$ to each node in $x$ |
| Inference | $\text{isinstance}(x, t)$ | Returns true if $x$ is an instance of $t$, false otherwise |
| | $\text{has}(x, p)$ | Returns true if $p$ is a property of $x$, false otherwise |
| | $\text{equal}(x, x')$ | Returns true if $x$ equals $x'$, false otherwise |
| Inquiry | $\text{parent}(x)$ | Returns the parent node of $x$ |
| | $\text{path}(x)$ | Returns the path from the tree root to $x$ |
| | $\text{get}(x, l)$ | Returns the sub-node of $x$ which has path $l$ |
| | $\text{query}(x, \theta)$ | Returns a dict of $\langle \text{path, value} \rangle$ pairs which contains all sub-nodes of $x$ satisfying predicate $\theta$ |
| Replication | $\text{clone}(x)$ | Returns a symbolic copy of $x$ |

## A.4 Materializing a child program from an abstract child program

As we decouple the search algorithm from the search space and child program by introducing the *abstract search space* and *abstract child program*, we need to materialize the abstract child program into a concrete child program based on the search space. Algorithm 1 illustrates this process, which recursively merges the hyper values from the search space and the numeric choices from the abstract child program. For a continuous or discrete hyper value, the value of choice is the final value to be

assigned to its target node in the tree, while for a categorical hyper value, the value of choice is the index of the selected candidate.

---

**Algorithm 1:** `materialize`

---

**Input:** $search\_space$, $abstract\_child\_program$
**Output:** $child\_program$

---

**if** `isinstance`($search\_space$, `Choice`) **then**
    $xcs \leftarrow ()$
    **forall** $c \in$ `child_space`($search\_space$) **do**
        $dc \leftarrow$ `child_value`($abstract\_child\_program$, `path`($c$))
        $xc \leftarrow$ `materialize`($c, dc$)
        `append`($xcs, xc$)
    $child\_program \leftarrow xcs[$`value_of`($abstract\_child\_program$)$]$
**else**
    $child\_program \leftarrow$ `value_of`($abstract\_child\_program$)
**return** $child\_program$

---

### A.5 Sampling child programs from a search space

Sampling a child program from a search space can be described as a process in which 1) the search algorithm proposes an abstract child program, and 2) the search space materializes the abstract child program into a concrete program. Before the process starts, an abstract search space will be obtained from the search space for setting up the search algorithm. This process is described by Algorithm 2.

---

**Algorithm 2:** `sample`

---

**Input:** $search\_space$, $search\_algorithm$
**Output:** `Iterator`($\langle child\_program, feedback\_for\_child \rangle$)

---

`setup`(search_algorithm, `abstract_search_space`(search_space))
**while** $true$ **do**
    $abstract\_child\_program \leftarrow$ `propose`($search\_algorithm$)
    $child\_program \leftarrow$ `materialize`($search\_space, abstract\_child\_program$)
    $feedback\_for\_child \leftarrow$
     `partial_bind`(`feedback`, $search\_algorithm, abstract\_child\_program$)
    **yield** $\langle child\_program, feedback\_for\_child \rangle$

---

# B More on PyGlove

In this section, we will map the concepts from our method into PyGlove programs, to illustrate how a regular Python program is made symbolic programmable, turned into a search space, and then optimized in a search flow. At the end of this section, we provide an example of enabling NAS for an existing Tensorflow-based MNIST program.

## B.1 Symbolize a child program

### B.1.1 Symbolize classes

A symbolic class can be converted from a regular Python class using the `@symbolize` decorator, or can be created on-the-fly without modifying the original class. The `symbolize` decorator creates a class on-the-fly by multi-inheriting the symbolic Object base class and the user class. The resulting class therefore possesses the capabilities of both parents. Figure 1 shows an code example of symbolizing existing/new classes.

```python
import pyglove as pg
import tensorflow as tf

# Symbolizing preexisting keras layers into symbolic
# classes without modifying original classes.
Conv2D = pg.symbolize(tf.keras.layers.Conv2D)
Dense = pg.symbolize(tf.keras.layers.Dense)
Sequential = pg.symbolize(tf.keras.Sequential)

# Symbolizing a newly created class with constraints.
@pg.symbolize([
  ('learning_rate', pg.typing.Float(min_value=0)),
  ('steps', pg.typing.Int(min_value=1))
])
class CosineDecay(object):

  def __init__(self, learning_rate, steps):
    self.learning_rate = learning_rate
    self.steps = steps

  def __call__(self, current_step):
    return (tf.cos(np.pi * current_step / self.steps)
            * self.learning_rate)
```

Figure 1: Symbolizing existing classes and new classes.

**Using symbolic constraints**   Constraints which validate new values during object construction or upon modification can be optionally provided when using the `@symbolize` decorator. Symbolic constraints can greatly reduce human mistakes when a program is manipulated by other programs. It also make the program implementation more crisp: user can program against an argument as it claims to be without additional check.

**Recomputing internal states**   Symbolic objects may have internal states. The mutable programming model will only work when the internal states are consistent upon modification. When one or more hyper-parameters are modified through `rebind`, the object's state will be reset, and the object's constructor will be invoked (again) on the same instance. Moreover, the change propagates back from the current node to the root of the symbolic tree, allowing all impacted nodes to recompute states upon modification.

### B.1.2 Symbolize functions

**From function to functor**   Making functions symbolic programmable is trickier than for classes, for the following reasons: First: functions don't explicitly hold their parameters as member variables,

although functions' bound arguments are analogous to member variables in classes. Second: functions don't have the concept of inheritance, which is necessary to get access to the capabilities provided by the symbolic Object base class. To address these two issues, we introduce the concept of *functor*, which is a symbolic class with a `__call__` method; all the function arguments becoming the functor's hyper-parameters. Under the functor concept, we unify the representation and operations of classes and functions. Figure 2 shows that functions can be symbolized in the same way as we symbolize classes. Figure 3 shows how functors can be used with great flexibility in binding their hyper-parameters.

```python
@pg.symbolize
def random_augment(image, magnitude):
  return random_augment_impl(data, magnitude)

@pg.symbolize([
  ('model', pg.typing.Instance(Layer)),
  ('augment_policy', pg.typing.Callable(
     [pg.typing.Instance(tf.Tensor)],
     returns=pg.typing.Instance(tf.Tensor))),
  ('learning_schedule', pg.typing.Callable([
     pg.typing.Instance(tf.Tensor)]))
])
def train_model(model,
                augment_policy,
                learning_schedule):
  return train_model_impl(
    model, augment_policy, learning_schedule)
```

Figure 2: Decorator `symbolize` converts functions into functors. Since properties for functors are automatically added from function signature, constraints are optional. Nevertheless, users are encouraged to add constraints for functor properties for safety and productivity.

```python
model = Sequential(children=[
  Conv2D(filters=8, kernel_size=(3, 3)),
  Dense(units=10)
])

# Partial parameter binding, in which 'model' is missing.
trainer = train_model(
  augment_policy=random_augment(
    magnitude=8))

# Incremental parameter binding via assignment.
trainer.learning_schedule = CosineDecay(1e-5, 5000)

# Incremental parameter binding at call time.
accuracy1 = trainer(model=model)

# Call with overriding previously bound parameters.
accuracy2 = trainer(
  model=model,
  learning_schedule=CosineDecay(2e-4, 5000),
  override_args=True)
```

Figure 3: Functors can be used as objects, with a rich set of argument binding features.

**Partial and incremental argument binding**   Functor comes with a capability that allows arguments to be partially bound at construction time, incrementally bound via property assignment and at call time. We can even override a previously bound argument during the call to the functor.

## B.2 Operating symbolic values

Symbolic values can be operated as if they were plain data, including inference, inquiry, modification and replication. Figure 4 gives some examples to these operations.

```python
model = Sequential(children=[
  Conv2D(filters=8, kernel_size=(3, 3)),
  Dense(units=10)
])

# Partial parameter binding, in which 'model' is missing.
trainer = train_model(
  augment_policy=random_augment(
    magnitude=8))

# Inference.
assert isinstance(trainer, train_model)
assert isinstance(trainer.model, Layer)
assert trainer.model.children[1] == Dense(10)
assert trainer.model != Conv2D(16, (3, 3))

# Inquiry.
assert trainer.query('.*filters') == {
    'model.children[0].filters': 8
  }
assert trainer.query(where=(
  lambda v: isinstance(v, Dense))) == {
    'model.children[1]': Dense(units=10)
  }

# Modification.
assert trainer.rebind({
    'model.children[0].filters': 16,
    'model.children[1]': insert(Dense(20))
  }).model == Sequential([
    Conv2D(16, (3, 3)), Dense(20), Dense(10)
  ])

def conv_to_dense(k, v):
  return Dense(v.filters) if isinstance(v.Conv2D) else v
assert trainer.rebind(conv_to_dense) == (
  Sequential([Dense(16), Dense(20), Dense(10)]))

# Replication.
assert trainer.clone() == trainer
assert trainer.clone(deep=True) == trainer
trainer.save('trainer.json')
assert pg.load('trainer.json') == trainer
```

Figure 4: Example code for symbolic operations on inference, comparison, inquiry, modification, replication and serialization.

## B.3 Using PyGlove for search

### B.3.1 Creating search spaces

With the definition of functors `train_model` and `random_augment`, as well as the layer classes, we can create a search space by replacing concrete values with hyper values, illustrated in Figure 5.

```
hyper_trainer = train_model(
  model=Sequential(
    pg.manyof(k=3, candidates=[
        Conv2D(filters=pg.oneof([8, 16]),
              kernel_size=pg.oneof([(3, 3), (5, 5)])),
        Dense(units=pg.oneof([10, 20]))
    ], choices_distinct=False)),
  augment_policy=random_augment(
      magnitude=pg.oneof([3, 6, 9])),
  learning_schedule=CosineDecay(pg.floatv(1e-5, 1e-4), 5000))
```

Figure 5: An example of conditional search space for jointly searching the model architecture, data augment policy, and learning rate.

### B.3.2 Search: putting things together

With `hyper_trainer` as the search space, we can start a search by sampling concrete trainers from the search space with a search algorithm (e.g. `RegularizedEvolution` [2]). The `trainer` is a concrete instance of `train_model`, which can be invoked to return the validation accuracy on ImageNet. We use the validation accuracy as a reward to feedback to the search algorithm, illustrated in Figure 6.

```
for trainer, feedback in pg.sample(
    hyper_trainer, pg.generators.RegularizedEvolution(),
    partition_fn=None):
  reward = trainer()
  feedback(reward)
```

Figure 6: Creating a search flow from a search space and a search algorithm. We pass None to the search space partition function here as to optimize the whole search space.

### B.4 More on materialization of hyper values

Materialization of hyper values can take place either eagerly or in a late-bound fashion. In the former case, the hyper value evaluates to a concrete value within its range upon creation, and register the search space into a global context for the first run, which can be picked up by the search algorithm later to propose values for future runs. This conditional evaluation makes it possible to support the define-by-run style search space definition advocated by Optuna [3]. In the latter case, the search space will be inspected from the symbolic tree and the tree can be manipulated freely by the search algorithm before the program is executed.

```
def oneof(candidates, hints=None):
  """Oneof with optional eager execution."""
  choice = Choice(candidates, hints)
  if is_eager_mode():
    if is_apply_decisions():
      # Apply next decision from the global context.
      chosen_index = next_global_decision()
    else:
      # Collect the decision points when running
      # the program for the first time.
      add_global_decision_point(choice)
      chosen_index = 0
    choice = candidates[chosen_index]
  return choice
```

Figure 7: Eagerly evaluation of hyper values.

The advantage of eager evaluation is that one can drop AutoML into a new ML program with minimal code changes. Users do not need to explicitly define the hyper-parameters to search. Instead, we can automatically identify them by executing the user's code before the start of the search. On the other hand, scattered searchable hyper-parameters makes it hard or error-prone to modify search space over many files, especially when we want to explore multiple search spaces.

Meanwhile, conditional search spaces require special handling. Define-by-run semantics typically do not provide enough information for us to recognize hierarchical search spaces. For instance, it is difficult to distinguish between `oneof([oneof([1, 2]), 1])` and `oneof([1, 2]) + oneof([3, 4])`. In PyGlove, we solve this problem by using a lambda function with zero-argument which returns the candidate: `oneof([lambda:oneof([1, 2]), 1])`. In this case, the outer `oneof` will instantiate the inner `oneof`, making it possible to capture the hierarchy of the hyper value structure.

While eagerly evaluation of hyper values seems to override the mechanism of symbolic manipulation, it is not so for PyGlove: Under eager mode, PyGlove runs the user program once to collect the symbolic objects (like the hyper values) along the program flow, so we can access these objects, manipulate them and inject them back into the program for future runs. As a result, eagerly evaluation can be regarded as an interface for PyGlove to inspect and manipulate the implicit symbolic objects created during program execution.

## B.5 Example: Neural Architecture Search on MNIST

This section shows a complete example of dropping PyGlove into an existing ML program as to enable NAS. Added code is highlighted with a light-yellow background.

```
"""NAS on MNIST.

This is a basic working ML program which does NAS on MNIST.
The code is modified from the tf.keras tutorial here:
https://www.tensorflow.org/tutorials/keras/classification

(The tutorial uses Fashion-MNIST,
but we just use "regular" MNIST for these tutorials.)

"""

from absl import app
from absl import flags
import numpy as np
import pyglove as pg
import tensorflow as tf

flags.DEFINE_integer(
    'max_trials', 10, 'Number of max trials for tuning.')

flags.DEFINE_integer(
    'num_epochs', 10, 'Number of epochs to train for each trail.')

FLAGS = flags.FLAGS

def download_and_prep_data():
  """Download dataset and scale to [0, 1].

  Returns:
    tr_x: Training data.
    tr_y: Training labels.
    te_x: Testing data.
    te_y: Testing labels.
  """
  mnist_dataset = tf.keras.datasets.mnist
  (tr_x, tr_y), (te_x, te_y) = mnist_dataset.load_data()
  tr_x = tr_x / 255.0
  te_x = te_x / 255.0
```

```python
    return tr_x, tr_y, te_x, te_y

# Create symbolized Keras layers classes.}
Conv2D = pg.symbolize(tf.keras.layers.Conv2D)

Dense = pg.symbolize(tf.keras.layers.Dense)

Sequential = pg.symbolize(tf.keras.Sequential)

def model_builder():
  """Model search space."""
  return Sequential(pg.oneof([
    # Model family 1: only dense layers.
    [
        tf.keras.layers.Flatten(),
        Dense(pg.oneof([64, 128]), pg.oneof(['relu', 'sigmoid']))
    ],
    # Model family 2: conv net.
    [
        tf.keras.layers.Lambda(lambda x:  tf.reshape(x, (-1, 28, 28, 1))),
        Conv2D(pg.oneof([64, 128]), pg.oneof([(3, 3), (5, 5)]) ,
             activation=pg.oneof(['relu', 'sigmoid'])) ,
        tf.keras.layers.Flatten()
    ]]) + [tf.keras.layers.Dense(10, activation='softmax')])

def train_and_eval(model, input_data, num_epochs=10):
  """Returns model accuracy after train and evaluation.

  Args:
    model: A Keras model.
    input_data: A tuple of (training features, training_labels,
      test features, test labels) as input data.
    num_epochs: Number of epochs to train model.

  Returns:
    Accuracy on test split.
  """
  tr_x, tr_y, te_x, te_y = input_data
  model.compile(optimizer='adam',
              loss='sparse_categorical_crossentropy',
              metrics=['accuracy'])

  model.fit(tr_x, tr_y, epochs=num_epochs)
  _, test_acc = model.evaluate(te_x, te_y, verbose=2)
  return test_acc

def search(max_trials, num_epochs):
  """Search MNIST model via PPO.

  Args:
    max_trials: Max trials to search.
    num_epochs: Number of epochs to train individual trial.
  """
  results = []
  input_data = download_and_prep_data()
  for i, (model, feedback) in enumerate(pg.sample(
      model_builder(), pg.generators.PPO(), max_trials)):
```

```python
    test_acc = train_and_eval(model, input_data, num_epochs)
    results.append((i, test_acc))
    feedback(test_acc)

  # Print best results.
  top_results = sorted(results, key=lambda x:  x[2], reverse=True)
  for i, (trial_id, test_acc) in enumerate(top_results[:10]):
    print('{0:2d} - trial {1:2d} ({2:.3f})'.format(i + 1, trial_id, test_acc))

def main(argv):
  """Program entrypoint."""
  if len(argv) > 1:
    raise app.UsageError('Too many command-line arguments.')
  search(FLAGS.max_trials, FLAGS.num_epochs)

if __name__ == '__main__':
  app.run(main)
```

# C More on case studies

This section describes the experiment details for our case studies in the paper.

## C.1 Search spaces and search algorithms exploration

### C.1.1 Experiment setup

Table 3: Hyper-parameters for training MobileNetV2 and searched models.

| Name | Value |
|------|-------|
| Image size | 224 * 224 |
| Pre-processing | ResNet preprocessing |
| Training epochs | 360 |
| Batch size | 4096 |
| Optimizer | RMSProp: momentum 0.9, decay 0.9, epsilon 1.0 |
| Learning schedule | Cosine decay with peak learning rate 2.64, with 5 epochs linear warmup at the beginning. |
| L2 | 2e-5 |
| Dropout rate | 0.15 |
| Batch normalization | momentum 0.99, epsilon 0.001 |

Table 4: Definition of search spaces for exploration.

| Search space | Hyper-parameters |
|--------------|------------------|
| $\mathcal{S}_1$ | Search the kernel sizes with candidates [(3, 3), (5, 5), (7, 7)] and expansion ratios with candidates [3, 6] for the inverted bottleneck units in MobileNetV2; can remove layers from each block. |
| $\mathcal{S}_2$ | Search output filters of MobileNetV2 with multipliers [0.5, 0.625, 0.75, 1.0, 1.25, 1.5, 2.0] |
| $\mathcal{S}_3$ | Combine $\mathcal{S}_1$ and $\mathcal{S}_2$ |

Table 5: Search algorithm setup. We uses TuNAS absolute reward function with exponent=-0.1.

| Search algorithm | Configuration |
|------------------|---------------|
| Random Search [4] | 100 trials, each trial trains for 90 epochs, rejection threshold for MAdds: $\pm$ 6M |
| Bayesian Optimization [5] | 100 trials, each trial trains for 90 epochs rejection threshold for MAdds: $\pm$ 30M (We use a larger rejection ratio for Bayesian to limit the rejection rate, since our infra will take the rewards from rejected trials) |
| TuNAS [6] | Search for 90 epochs, with RL learning rate set to 0 for first 1/4 of training. The search cost for $\mathcal{S}_1$ and $\mathcal{S}_2$ is about 4x of static model training, and the cost for $\mathcal{S}_3$ about 8x of static model training. |

### C.1.2  Creating search spaces

In the Result section, we demonstrated 3 search spaces created from MobileNetV2 [7]. Figure 8-10 show the code for converting the static model into search spaces $\mathcal{S}_1$, $\mathcal{S}_2$ and $\mathcal{S}_3$.

```python
import pyglove as pg

# Get the first inverted bottleneck.
r = model.query(lambda x: isinstance(x, InvertedBottleneck))
r = next(iter(r.values()))

def hyper_inverted_bottleneck(
  kernel_size_list, expansion_ratio_list, add_zeros=False):
    return pg.oneof([
      r.clone().rebind(kernel_size=k, expansion_ratio=e)
      for k in kernel_size_list
      for e in expansion_ratio_list
    ] + ([Zeros()] if add_zeros else []))

def relax_ops(k, v, p):
  if not k or k.key != 'op':
    return v
  # Check if the layer of current operation is the
  # first layer of current block.
  if k.parent.key == 0:
    if k == 'blocks[0].layers[0].op':
      return hyper_inverted_bottleneck(
        [(3, 3), (5, 5), (7, 7)], [1])
    else:
      return hyper_inverted_bottleneck(
        [(3, 3), (5, 5), (7, 7)], [3, 6])
  else:
    return hyper_inverted_bottleneck(
      [(3, 3), (5, 5), (7, 7)], [3, 6], True)

mobile_s2 = mobilenet_v2.clone().rebind(relax_ops)
```

Figure 8: Added code for converting MobileNetV2 into a search space ($\mathcal{S}_1$) that tunes the kernel size and expansion ratio in all inverted bottleneck units.

```python
def relax_filters(k, v, p):
  if isinstance(p, InvertedBottleNeck) and k == 'filters':
    scaled_values = sorted(set([
      layers.scale_filters(v, x)()
      for x in [0.5, 0.625, 0.75, 1.0, 1.25, 1.5, 2.0]]))
    if len(scaled_values) == 1:
      return scaled_values[0]
    return pg.oneof(scaled_values)
  return v
mobile_s1 = mobilenet_v2.clone().rebind(relax_filters)
```

Figure 9: Added code for converting MobileNetV2 into a search space ($\mathcal{S}_2$) that tunes the channel size in all inverted bottleneck units.

```python
mobile_s3 = mobilenet_v2.clone().rebind([relax_filters,
                                         relax_ops])
```

Figure 10: Applying transform functions from $\mathcal{S}_1$ and $\mathcal{S}_2$ to create $\mathcal{S}_3$.

## C.2  Search flow exploration

In the case study, we explored 3 search flows for optimizing NAS-Bench-101. Here we include the code for the factorized and hybrid search since the standard search is already discussed in Section B.3.2.

### C.2.1  Factorized search

For the *factorized* search, we optimize the nodes in the outer loop and the edges in the inner loop. Each example in the outer loop is a search space of edges with a fixed node setting. Each example in the inner loop is a fixed model architecture. The reward for the outer loop is computed as the average of top 5 rewards from the inner loop.

```
def factorized_search(search_space):
  # Optimize the ops in the outer loop.
  # Each example in the outer loop is an edge sub-space with fixed
  # ops. `partition_fn` is used to create a sub-space by selecting
  # op hyper values only.
  best_example, best_reward = None, None
  for edge_space, ops_feedback in pg.sample(
      search_space, RegularizedEvolution(),
      trials=300, partition_fn=lambda v: v.hints == OP_HINT):
    # Optimize the edges in the inner loop.
    # Each reward computed in the inner loop
    # is for an edge setting relative to
    # the node setting decided in the outer loop.
    rewards = []
    for example, edges_feedback in pg.sample(
        edge_space, RegularizedEvolution(), trials=20):
      reward = nasbench.get_reward(example)
      edges_feedback(reward)
      rewards.append(reward)
      if best_reward is None or best_reward < reward:
        best_example, best_reward = example, reward
    ops_feedback(top5_average(rewards))
  return best_example
```

Figure 11: A factorized search that optimizes the nodes in the outer loop and the edges in the inner loop.

### C.2.2 Hybrid search

For the *hybrid* search, we use the first half of the budget to optimize the nodes using the same search flow illustrated in Section C.2.1 , then we use the other half of the budget to further optimize the edges with the best nodes found in the prior phase.

```python
def hybrid_search(search_space):
  # Phase 1: search for the best ops with sampled edges.
  # Each example in the outer loop is an edge sub-space with fixed
  # ops. 'partition_fn' is used to create a sub-space by selecting
  # op hyper values only.
  ops_attempts = []
  for edge_space, ops_feedback in pg.sample(
      search_space, RegularizedEvolution(),
      trials=150, partition_fn=lambda v: v.hints == OP_HINT):
    rewards = []
    algo = RegularizedEvolution()
    for example, edges_feedback in pg.sample(edge_space,
                                             algo, trials=20):
      reward = nasbench.get_reward(example)
      edges_feedback(reward)
      rewards.append(reward)
    ops_reward = top5_average(rewards)
    ops_attempts.append((edge_space, ops_reward, algo))
    ops_feedback(ops_reward)

  # Phase 2: Continue search the best edge sub-space
  # with best ops found.
  edge_space, _, edge_algo = sorted(
      ops_attempts, key=lambda x: x[1], reverse=True)[0]

  best_example, best_reward = None, None
  for example, edges_feedback in pg.sample(edge_space,
                                           edge_algo, 150 * 20):
    reward = nasbench.get_reward(example)
    edges_feedback(reward)
    if best_reward is None or best_reward < reward:
      best_example, best_example = example, reward
  return best_example
```

Figure 12: A hybrid search that optimizes the nodes with a factorized search in the first phase, and optimize the edges based on the best nodes found in the second phase.