[Reviews · NeurIPS 2020]

Review 1

Summary and Contributions: This paper introduces PyGlove—a Python library for AutoML. PyGlove makes implementing AutoML easier by decoupling the search space, search algorithm, and child programs. The decoupling is realized by symbolic programming. The search space and programs are represented as symbolic trees, which is manipulated by the search algorithm. The authors present use cases on ImageNet and NAS-Bench-101, demonstrating that users can implement complex search flows easily with PyGlove.

Strengths: + The proposed PyGove library could be a useful tool for AutoML researchers and practitioners. In existing tools for AutoML, the search space is often coupled with the search algorithm, making it difficult to change either of them. In contrast, PyGlove uses symbolic programming to decouple the search space and search algorithm, which makes it easier to experiment with different prototypes. + Use cases on ImageNet and NAS-Bench-101 demonstrate that we can implement different NAS algorithms in PyGlove easily. The high-level search strategy can be described by only a few lines of code. + The paper is well-written and easy to follow.

Weaknesses: - The language for specifying the search space may not be expressive enough to capture a wide range of search space designs. Currently, it has "floatv" for a floating-point range, "intv" for an integer range, "oneof" for choosing one value from a candidate list, and "permutate" for generating permutations of modules. It would be great to discuss whether these primitives can cover search spaces designed in existing NAS literature. - It would be more convincing if the authors include use cases where they replicate prior work in NAS using PyGlove. If it's possible to do so with very few lines of code, this is strong evidence for the usefulness of PyGlove.

Correctness: I don't see any significant flaws in methodology.

Clarity: Yes.

Relation to Prior Work: Yes.

Reproducibility: Yes

Additional Feedback: ----------------------- Post-rebuttal Update I have read the rebuttal and other reviews. The authors have addressed my questions to a reasonable extent. I would recommend accept for this paper.


Review 2

Summary and Contributions: The paper introduces an AutoML library that tries to find its own sweet spot in the large ecosystem of newly minted AutoML libraries. The paper introduces a symbolic frontend to build neural network models, with simple fundamental constructs that provide choice insertions. Unlike all other packages that I have seen and reviewed, such as Keras Tuner, NNI, AutoGluon, Optuna (btw reference missing to Optuna, you should consider adding), this paper introduces something innovative and elegant. All these other packages consistently suffer from the code of the model definition getting ugly and unweildy really quickly when you have to introduce model structure searches, and when there's interaction between structure searches and size searches. In this paper, the authors cleanly separate model structure definitions from each layer's hyperparameter choices. They do this by introducing their own meta-layers which are put together in a symbolic frontend (ironically, TensorFlow v1 would've been a good out of the box candidate to be fitting for this, but v2 doesn't allow the same). Each of these meta-layers can be structurally permuted, as well as swapped with a dictionary of layers, using a simple meta-programming API that has some convenient pre-defined constructs. The general interface for interacting in different modes of search (weight sharing, layer swapping, width changing, etc.) is convincingly good, and articulated well. The results section of this package is a bit zero-signal, but at the same time it's really hard to write a convincing results section for frontend innovation. So, I'm completely discounting this as **not relevant for the type of paper**. The authors are honest in the broader impact that AutoML is going to release waaaaaay more CO2 than traditional hand-optimization.

Strengths: Despite entering a crowded space, the paper does a remarkable job of solving a novel problem of releasing the tension between writing the model code in a non-ugly manner and inserting the automl search parameters, especially in the case of searching among new layers and layer orders.

Weaknesses: The paper is a software paper, and doesn't fit a "traditional" NeurIPS style of "results" section being about how the authors deserve a prize for state of the art. It's not a weakness, but it can be perceived so. I want to express it as a paper submitted potentially to the limitations of the venue that it's submitted to.

Correctness: The paper seems to be generally correct and the methodology for results seems fine.

Clarity: The paper is reasonably well-written and easy to follow. One big gripe I have is that the authors don't define "cost" in the Results section. I am **assuming** that it might be program runtime, but it's not clear. The line they have is: > The unit cost for search and training is defined as the computation cost to train a MobileNetV2-like model on ImageNet for 360 epochs. It could literally be anything, like the cost of buying the program from someone, the memory usage of the program, the environmental cost of CO2 emissions, and a thousand other things I can make up. Define the cost more precisely.

Relation to Prior Work: Prior work is discussed comprehensively and fairly. I went through the documentation of all the discussed prior work (and additional libs such as Optuna) to make sure I understand the fairness of claims.

Reproducibility: Yes

Additional Feedback:


Review 3

Summary and Contributions: This paper presents PyGlove, a Python library for AutoML. The problem PyGlove aims to solve is managing inconsistencies between the search space and search algorithm in AutoML when one or the other is modified. It introduces a symbolic representation of Python objects that is able to express neural architectures and training procedures. These representations can be searched and modified automatically, making it easier to decouple the search space and search algorithm.

Strengths: The primary strength is that’s they have developed a useful tool that hits a sweet spot in the design space of random search based program synthesis methods for machine learning. Their approach can be applied to essentially parameterize existing deep network architectures, since introducing a choice amounts to only adding small annotations to existing python code. Ultimately, changes to the search space or search algorithm can be done with small modifications to the existing code.

Weaknesses: One weakness is the exposition and relation to prior work. Either the authors are not aware of how their approach sits within the broader framework of programming languages or synthesis, or they have chosen their language to hide it to make the work more approachable. For instance, they introduce their objects as symbolic trees, which are simply abstract syntax trees. In fact, all programs are represented at some point as syntax trees. Second, most of their tool/language design could be summarized as adding some kind of non deterministic/parametric choice into the programming language, so that the search mechanism, for instance, can choose between 16 or 32 layers. This has been explored in many areas in PL. It’s extension to ML does not introduce anything particularly new, but that it is not to say it is not useful. The evaluation is mostly not inline with the objectives of the paper. The paper does not suggest new ways of searching for programs, it is a tool designed to make it easier to change search space algorithms independently from changes to the search spaces themselves. Consequently, the performance of the resulting searches is not really relevant, and the paper should spend much more ink expanding on the how easy it is to do these search space / search algorithm changes compared to other approaches.

Correctness: There are few claims to be assessed for correctness, but the method appears sound.

Clarity: Clear.

Relation to Prior Work: Relation to existing AutoML work is clear, but severely lacking in connections to PL methods, which have long explored this.

Reproducibility: Yes

Additional Feedback: Provide the grammar in the main text. ------------------------------ UPDATE ------------- After reading the other reviews it has become more apparent to me how useful this approach is to the AutoML community both as a tool and methodology. As a result I'll raise my score slightly. As I mention in my review you should draw on the connections between this work and the large body of existing work in the PL literature.

[Author Response · NeurIPS 2020]

We thank all reviewers for their valuable feedback and suggestions. As a submission to NeurIPS, we hope our work will be useful for the large AutoML audience present in this conference. Here we will first address a shared concern on how to make the "result" section more useful, then discuss specific concerns from individual reviewers.

**R1, R3, R4:** Improving the usefulness of our experimental evaluation.

All three reviewers noted that PyGlove's key innovations were on the framework side rather than on the NAS algorithms side, and expressed concerns that the empirical evaluation of search results was largely orthogonal to the paper's main contributions. We agree with the reviewers' concerns. To address them, we will make the following changes: 1) mention that we have also successfully reproduced a wide variety of popular NAS papers with diverse search spaces and algorithms, as evidence for PyGlove's expressive power: NASBench, MNASNet, NAS-FPN, SpineNet, ProxylessNAS, TuNAS. 2) clarify that the "result" section should be a "case study" section, which demonstrates that PyGlove can easily try out different search spaces and algorithms, and quantifies the reduction in engineering effort.

**R1:** *"The language for specifying the search space may not be expressive enough..."* **and** *"... include use cases where they replicate prior work in NAS using PyGlove..."*

We agree that expressiveness is important. PyGlove can express very complex search spaces, in particular 1) conditional search spaces, representable by nested "oneof" or "manyof"; 2) interdependent parameters, which can be expressed by high-order symbolic values, discussed in Appendix A.4. The expressiveness is further verified by our reproduced NAS papers listed above, which will be added. For clarity, we will also add more details to the "search space" sub-section.

**R3**: *"Unlike all other packages that I have seen and reviewed, such as Keras Tuner, NNI, AutoGluon, Optuna (btw reference missing to Optuna, you should consider adding), this paper introduces something innovative and elegant."*

Thanks for going through an extensive list of AutoML toolkits in verifying the fairness of our claim and writing an in-depth analysis of PyGlove's method. We will add a discussion of Optuna in the revision.

**R3**: *"Define the cost more precisely."*

We will define the training/search cost more precisely using GPU hours.

**R4:** *"One weakness is the exposition and relation to prior work...For instance, they introduce their objects as symbolic trees, which are simply abstract syntax trees..."*

Thanks for bringing up the relation to PL/PS. To our best knowledge, our work can be related to symbolic programming, non-deterministic programming, functional programming, and program synthesis. We will discuss their connections in more detail with the following references. We would be happy to add more based on further reviewer suggestions.

- GJ Sussman, MIT, 2007, Building robust systems an essay
- Abelson, H. and Sussman, G.J., 1996. Structure and interpretation of computer programs
- Andre, D., and Russell, S. J. 2002. State abstraction in programmable reinforcement learning
- H. Søndergaard, P. Sestoft, 1992, Non-determinism in functional languages
- A Solar-Lezama, The sketching approach to program synthesis, Springer, 2009
- S. Gulwani. Synthesis of loop-free programs. In PLDI, pp. 62–73, 2011

Regarding the terminology, we preferred "symbolic tree" over "AST" for two reasons. First, as R4 suggested, "symbolic tree" was more approachable for people in the ML community. Second, the symbolic tree is declared by the user using decorators and serves to represent high-level program constructs, which is different from the AST that represents all the syntactic structures for the program. For example, the full Python AST contains information about objects' class methods, whereas our symbolic representation does not.

**R4**: *"Second, most of their tool/language design could be summarized as adding some kind of non deterministic/parametric choice ... It's extension to ML does not introduce anything particularly new ..."*

We agree with R4 that symbolic programming and non-deterministic programming are well-studied topics in the PL community. However, we would like to emphasize that this work is the first to introduce such concepts to AutoML to significantly reduce engineering effort, which is a novel and useful contribution. For example, PyGlove leverages symbolic manipulation to decouple the search algorithm, search space and child program, which enabled us to unify the interface among search methods with and without weight sharing. To enable symbolic programming in Python, PyGlove implements an object model for maintaining the consistency of program state during symbolic manipulation.

**R4** *"Provide the grammar in the main text"*

We understand the "grammar" here as a reference to the formal definition of the search space specification. We will revise current Appendix Table 3 into a formal definition, and add it to the "search space" sub-section.

[Meta-Review · NeurIPS 2020]

The reviewers generally agree that the design choices of this framework for AutoML are judicious and hit a "sweet spot". This combination of language/tooling design is of great value to expose to large swathes of the NeurIPS community. The rebuttal persuasively addresses the reviewers' concerns about the evaluation and utility of this proposal, and the response to R4 is also reassuring. We look forward to the authors' final version of the paper, incorporating the proposed improvements.